# NEURON AS AN AGENT

## ABSTRACT

Existing multi-agent reinforcement learning (MARL) communication methods have relied on a trusted third party (TTP) to distribute reward to agents, leaving them inapplicable in peer-to-peer environments. This paper proposes reward distribution using *Neuron as an Agent* (NaaA) in MARL without a TTP with two key ideas: (i) inter-agent reward distribution and (ii) auction theory. Auction theory is introduced because inter-agent reward distribution is insufficient for optimization. Agents in NaaA maximize their profits (the difference between reward and cost) and, as a theoretical result, the auction mechanism is shown to have agents autonomously evaluate counterfactual returns as the values of other agents. NaaA enables representation trades in peer-to-peer environments, ultimately regarding unit in neural networks as agents. Finally, numerical experiments (a single-agent environment from OpenAI Gym and a multi-agent environment from ViZDoom) confirm that NaaA framework optimization leads to better performance in reinforcement learning.

## 1 INTRODUCTION

Now that intelligence with reinforcement learning has been shown to be superior to humans (Tesauro, 1995; Mnih et al., 2015; Silver et al., 2016), reinforcement learning is expected to expand into industry applications such as stock trading, autonomous cars, smart grids, and IoT. In future industrial applications, numerous companies will own agents used to improve their revenues. Such a situation can be considered as each agent independently solving the problems of a partially observed Markov decision process (POMDP).

These company agents are designed to independently maximize their own rewards; however, if such agents could exchange information, the overall revenue of stakeholders would increase. As each agent has limited visibility of the environment, information exchanges between agents would be helpful in solving their independent tasks. Thus, this paper aimed to actualize a society where stakeholders with conflicting interests willingly trade information.

We regard the situation as communication in multi-agent reinforcement learning (MARL). Multi-agent communication is addressed by several methods such as R/DIAL (Foerster et al., 2016) and CommNet (Sukhbaatar et al., 2016). CommNet is a state-of-the-art MARL method that considers communication between agents and features learning among agents with backpropagation.

When considering conditions in MARL such that different stakeholders have created different agents that communicate with each other, an reward distribution design (e.g., monetary payment) and a framework without a *trusted third party* (TTP) are required. TTP (Wu & Soo, 1999; Sandholm & Wang, 2002) is a neutral administrator that performs reward distributions for all participants, and is supposed implicitly by most existing literature regarding MARL (Agogino & Tumer, 2006; Foerster et al., 2016; Sukhbaatar et al., 2016). Although there is a requirement for TTP neutrality towards all participants, several peer-to-peer trade configurations, such as inter-industry and -country trades, cannot place TTP. If reward distribution is performed by an untrusted third party, the rewards for partial participants may be undesirably altered.

To the best of our knowledge, no existing literature discusses reward distributions in the configuration described above. Because CommNet assumes an environment that distributes a uniform reward to all the agents, if the distributed reward is in limited supply (such as money), it causes the *Tragedy of the Commons* (Lloyd, 1833), where the reward of contributing agents will be reduced due to the participation of free riders. Although there are several MARL methods for distributing rewards ac-

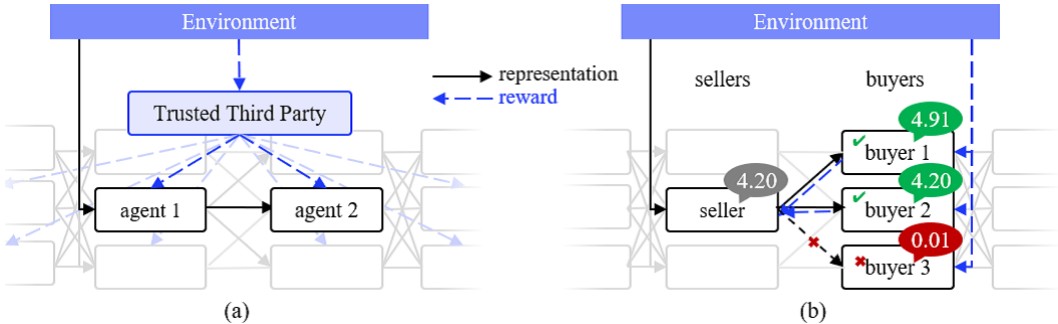

Figure 1: A schematic illustration of reward distribution models in MARL. **(a)** Centralized reward distribution model (Agogino & Tumer, 2006; Sukhbaatar et al., 2016; Foerster et al., 2016; 2017). They should suppose TTP to distribute the optimal reward to the agents. **(b)** Inter-agent reward distribution model (our model). Some agents receive reward from the environment directly, and redistribute to other agents. The idea to determine the optimal reward without TTP is playing auction game among the agents.

cording to agents' contribution such as QUICR (Agogino & Tumer, 2006) and COMA (Sukhbaatar et al., 2016), they suppose the existence of TTP and hence cannot be applied to the situation investigated here.

The proposed method, *Neuron as an Agent* (NaaA), extends CommNet to actualize reward distributions in MARL without TTP based on two key ideas: (i) inter-agent reward distribution and (ii) auction theory. Auction theory was introduced because inter-agent reward distributions were insufficient for optimization. Agents in NaaA maximize *profit*, the difference between their received rewards and the costs which they redistribute to other agents. If the framework is naively optimized, a trivial solution is obtained where agents reduce their costs to zero to maximize profits. Then, NaaA employs the auction theory in game design to prevent costs from dropping below their necessary level. As a theoretical result, we show that agents autonomously evaluate the *counterfactual return* as values of other agents. The counterfactual return is equal to the discounted cumulative sum of counterfactual reward (Agogino & Tumer, 2006) distributed by QUICR and COMA.

NaaA enables representation trades in peer-to-peer environments and, ultimately, regards neural network units as agents. As NaaA is capable of regarding units as agents without losing generality, this setting was utilized in the current study. The concept of the proposed method is illustrated in Figure 1.

An environment extending ViZDoom (Kempka et al., 2016), a POMDP environment, to MARL was used for the experiment. Two agents, a cameraman sending information and a main player defeating enemies with a gun, were placed in the environment. Results confirmed that the cameraman learned cooperative actions for sending information from dead angles (behind the main player) and outperformed CommNet in score.

Interestingly, NaaA can apply to single- and multi-agent settings, since it learns optimal topology between the units. *Adaptive DropConnect* (ADC), which combines DropConnect (Wan et al., 2013) (randomly masking topology) with an adaptive algorithm (which has a higher probability of pruning connections with lower counterfactual returns) was proposed as a further application for NaaA. Experimental classification and reinforcement learning task results showed ADC outperformed DropConnect.

The remainder of this paper is organized as follows. In the next section, we show the problem setting. Then, we show proposed method with two key ideas: inter-agent reward distribution and auction theory in Section 3. After related works are introduced in Section 4, the experimental results are shown in classification, single-agent RL and MARL in Section 5. Finally, a conclusion ends the paper.

## 2 PROBLEM DEFINITION

Suppose there is an $N$-agent system in an environment. The goal of this paper was to maximize the discounted cumulative reward the system obtained from the environment. This was calculated as:

$$G = \sum_{t=0}^{T} \gamma^t R_t^{\text{ex}}, \tag{1}$$

where $R_t^{\text{ex}}$ is a reward which the system obtains from the environment at $t$, and $\gamma \in [0,1]$ is the discount rate and $T$ is the terminal time.

Reward distribution is distributing $R_t$ to all the agents under the following constraint.

$$\sum_{i=1}^{N} R_{it} = R_t^{\text{ex}}, \tag{2}$$

where $R_{it}$ is a reward which is distributed to $i$-th agent at time $t$. For instance, in robot soccer, the environment give a reward 1 when an agent shoot a ball to the goal. Each agent should receive the reward along to their contribution.

In most of MARL communication methods, the policy of reward distribution is determined by a centralized agent. For example, QUICR (Agogino & Tumer, 2006) and COMA (Foerster et al., 2017) distribute $R_{it}$ according to counterfactal reward, difference of reward between an agent made an action and not. The value of counterfactual reward is calculated by centralized agent, called *trusted third party* (TTP).

In a peer-to-peer environment such as inter-industry and -country trade, they cannot place a TTP. Hence, another framework required to actualize reward distribution without TTP.

## 3 PROPOSED METHOD

The proposed method, *Neuron as an Agent* (NaaA), extends CommNet (Sukhbaatar et al., 2016) to actualize reward distributions in MARL without TTP based on two key ideas: (i) inter-agent reward distribution and (ii) auction theory. As we show in Section 3.3, NaaA actualizes that we can regard even a unit as an agent.

### 3.1 INTER-AGENT REWARD DISTRIBUTION

Some agents receive rewards from the environment directly $R_{it}^{\text{ex}}$, and they distribute these to other agents as incentives for giving precise information. Rewards are limited, so if an agent distributes $\rho$ rewards, the reward of that agents is reduced by $\rho$ to satisfy the constraint of Eq:(2). For this reason, agents other than a specified agent of interest can be considered a secondary environment for the agent giving rewards of $-\rho$ instead of an observation $x$. This secondary environment was termed the *internal environment*, whereas the original environment was called the *external environment*. Similarly to CommNet (Sukhbaatar et al., 2016), the communication protocol between agents was assumed to be a continuous quantity (such as a vector), the content of which could be trained by backpropagation.

A communication network among the agents is represented as a directed graph $\mathfrak{G} = (\mathcal{V}, \mathcal{E})$ between agents, where $\mathcal{V} = \{v_1, \ldots, v_N\}$ is a set of the agents and $\mathcal{E} \subset \mathcal{V}^2$ is a set of edges representing the connections between two agents. If $(v_i, v_j) \in \mathcal{E}$, then connection $v_i \rightarrow v_j$ holds, indicating that $v_j$ observes the representation of $v_i$. Here, the representation of agent $v_i$ at time $t$ was denoted as $x_{it} \in \mathbb{R}$. Additionally, the set of agents that agent $i$ connects to was designated to be $N_i^{\text{out}} = \{j | (v_i, v_j) \in \mathcal{E}\}$ and the set of agents that agent $i$ is connected from was $N_i^{\text{in}} = \{j | (v_j, v_i) \in \mathcal{E}\}$. Furthermore, $N_i = N_i^{\text{in}} \cup N_i^{\text{out}}$. The following assumptions were added to the $v_i$ characteristics:

- N1: (Selfishness) The utility each agent $v_i$ wants to maximize is its own return (cumulative discounted reward): $G_{it} = \sum_{k=0}^{T} \gamma^k R_{i,t+k}$.
- N2: (Conservation) The summation of internal rewards over all $\mathcal{V}$ equals 0. Hence, the summation of rewards which $\mathcal{V}$ (receive both internal and external environment $R_{it}$) are equivalent

to the reward $R_t^{\text{ex}}$, which the entire multi-agent system receives from the external environment: $\sum_{i=1}^{N} R_{it} = \sum_{i=1}^{N} R_{it}^{\text{ex}} = R_t^{\text{ex}}$.

N3: (Trade) An agent $v_i$ will receive an internal reward $\rho_{jit}$ from $v_j \in \mathcal{V}$ in exchange for an representation signal $x_i$ before transferring this signal to the agent. Simultaneously, $\rho_{jit}$ will be subtracted from the reward of $v_j$.

N4: (NOOP) $v_i$ can select NOOP (no operation), for which the return is $\delta > 0$, as an action. In NOOP, the agent inputs and outputs nothing.

The social welfare function (total utility of the agents) $G^{\text{all}}$ is equivalent to the objective function $G$. That is,

$$G^{\text{all}} = \sum_{i=1}^{N} G_{it} = \sum_{i=1}^{N} \left[ \sum_{k=0}^{T} \gamma^t R_{it} \right] = \sum_{k=0}^{T} \left[ \gamma^t \sum_{i=1}^{N} R_{it} \right]. \tag{3}$$

From N2, $G^{\text{all}} = G$ holds.

### 3.1.1 DISCOUNTED CUMULATIVE PROFIT MAXIMIZATION

From N3, the reward $R_{it}$ received by $v_i$ at $t$ can be written as:

$$R_{it} = R_{it}^{\text{ex}} + \sum_{j \in N_i^{\text{out}}} \rho_{jit} - \sum_{j \in N_i^{\text{in}}} \rho_{ijt}, \tag{4}$$

which can be divided into positive and negative terms, where the former is defined as revenue, and the latter as cost. These are respectively denoted as $r_{it} = R_{it}^{\text{ex}} + \sum_{j \in N_i^{\text{out}}} \rho_{jit}$, $c_{it} = \sum_{j \in N_i^{\text{in}}} \rho_{ijt}$. Here, $R_{it}$ represents *profit*, difference between revenue and cost.

The agent $v_i$ maximizes its cumulative discounted profit, $G_{it}$, represented as:

$$G_{it} = \sum_{k=0}^{T} \gamma^k R_{i,t+k} = \sum_{k=0}^{T} \gamma^k (r_{i,t+k} - c_{i,t+k}) = r_t - c_t + \gamma G_{i,t+1}. \tag{5}$$

$G_{it}$ could not be observed until the end of an episode (the final time). Because predictions based on current values were needed to select optimal actions, $G_{it}$ was approximated with the value function $V_i^{\pi_i}(s_{it}) = \mathbb{E}_{\pi_i}[G_{it} \mid s_{it}]$ where $s_{it}$ is a observation for $i$-th agent at time $t$. Under these conditions, the following equation holds:

$$V_i^{\pi_i}(s_{it}) = r_{it} - c_{it} + \gamma V_i^{\pi_i}(s_{i,t+1}), \tag{6}$$

Thus, herein, we only consider maximization of revenue, the value function, and cost minimization. The inequality $R_{it} > 0$ (i.e., $r_{it} > c_{it}$) indicates that the agent in question gave additional value to the obtained data. The agent selected the NOOP action because $V_i^{\pi_i}(s_{it}) \leq 0 < \delta$ if $R_{it} \leq 0$ for all $t$ (because of N4).

### 3.1.2 SOCIAL DILEMMA

If we directly optimize Eq:(6), a trivial solution is obtained in which the internal rewards converge at 0, and all agents (excepting agents which directly receive reward from the external environment) select NOOP as their action. This phenomenon occurs regardless of the network topology $\mathfrak{G}$, as no nodes are incentivized to send payments $\rho_{ijt}$ to other agents. With this in mind, multi-agent systems must select actions with no information, achieving the equivalent of taking random actions. For that reason, the total external reward $R_t^{\text{ex}}$ shrinks markedly. This phenomenon also known as social dilemma in MARL, which is caused from a problem that each agent does not evaluate other agents' value truthfully to maximize their own profit. We are trying to solve this problem with auction theory in Section 3.2.

### 3.2 AUCTION THEORY

To make the agents to evaluate other agents' value truthfully, the proposed objective function borrows its idea from the digital goods auction theory (Guruswami et al., 2005). In general, an auction theory is a part of mechanism design intended to unveil the true price of goods. Digital goods auctions are one mechanism developed from auction theory, specifically targeting goods that may be copied without cost such as digital books and music.

### 3.2.1 ENVY-FREE AUCTION

Although several variations of digital goods auctions exist, an envy-free auction (Guruswami et al., 2005) was used here because it required only a simple assumption: equivalent goods have a single simultaneous price. In NaaA, this can be represented by the following assumption:

N5: (Law of one price) If $\rho_{j_1,i,t}, \rho_{j_2,i,t} > 0$, then $\rho_{j_1,i,t} = \rho_{j_2,i,t}$.

The assumption above indicates that $\rho_{jit}$ takes either 0 or a positive value, depending on $i$ at an equal selected time $t$. Therefore, the positive side was named $v_i$'s *price*, and denoted as $q_{it}$.

The envy-free auction process is shown in the left section of Figure 2, displaying the negotiation process between one agent sending an representation (defined as a seller), and a group of agents buying the representation (defined as buyers). First, a buyer places a bid with the agent at a bidding price $b_{jit}$ (**1**). Next, the seller selects the optimal price $\hat{q}_{it}$ and allocates the representation (**2**). Payment occurs if $b_{ijt}$ exceeds $q_{jt}$. In this case, $\rho_{jit} = H(b_{jit} - q_{it})q_{it}$ holds where $H(\cdot)$ is a step function. For this transaction, the definition $g_{jit} = H(b_{jit} - q_{it})$ holds, and is named *allocation*. After allocation, buyers perform payment: $\rho_{jit} = g_{jit}\hat{q}_{it}$ (**3**). The seller sends the representation $x_i$ only to the allocated buyers (**4**). Buyers who do not receive the representation approximate $x_i$ with $\mathbb{E}_\pi[x_i]$. This negotiation is performed at each time step in reinforcement learning.

### 3.2.2 THEORETICAL ANALYSIS

The sections below discuss the revenue, cost, and value functions based on Eq:(6).

**Revenue**: The revenue of an agent is given as

$$r_{it} = \sum_{j \in N_i^{\text{out}}} g_{jit}q_{it} + R_{it}^{\text{ex}} = q_{it}d_{it} + R_{it}^{\text{ex}}, \tag{7}$$

where $d_{it} = \sum_{j \in N_i^{\text{out}}} g_{jit}$ is the *demand*, the number of agents for which the bidding price $b_{jit}$ is greater than or equal to $q_{it}$. Because $R_i^{\text{ex}}$ is independent of $q_{it}$, the optimal price $\hat{q}_{it}$ maximizing $r_{it}$ is given as:

$$\hat{q}_{it} = \underset{q \in [0,\infty)}{\operatorname{argmax}} \, q d_{it}(q). \tag{8}$$

The $r_{it}$ curve is shown on the right side of Figure 2.

**Cost**: Cost is defined as an internal reward that one agent pays to other agents. It is represented as:

$$c_{it} = \sum_{j \in N^{\text{in}}} g_{ijt}q_{jt} = \mathbf{g}_{it}^{\text{T}}\mathbf{q}_t, \tag{9}$$

where $\mathbf{g}_{it} = (g_{i1t}, \ldots, g_{iNt})^{\text{T}}$ and $\mathbf{q}_t = (q_{1t}, \ldots, q_{Nt})^{\text{T}}$. Although $c_{it}$ itself is minimized when $b_{ijt} = 0$, this represents a tradeoff with the following value function.

**Value Function**: The representation $x_{it}$ depends on input from the agents in $N_i^{in}$ affecting the bidding price from agents in $N_i^{out}$. If $b_{ijt}$ is minimized and $b_{ijt} = 0$, the representation purchase fails and the reward obtained by the seller agent from the connected buyer agents is lower in the future.

The effects of the value function were considered for both successful and unsuccessful $v_j$ purchasing cases. The value function was approximated as a linear function $\mathbf{g}_{it}$:

$$V_i^{\pi_i}(s_{i,t+1}) \approx \mathbf{o}_{it}^{\text{T}}\mathbf{g}_{it} + V_{i,t+1}^0, \tag{10}$$

where $\mathbf{o}_{it}$ is equivalent to the cumulative discount value of the counterfactual reward (Agogino & Tumer, 2006), it was named *counterfactual return*. As $V_{it}^0$ (a constant independent of $\mathbf{g}_{it}$) is equal to the value function when $v_i$ takes an action without observing $x_1, \ldots, x_N$.

The optimization problem is, therefore, presented below using a state-action value function for $i$-th agent $Q_i(\cdot, \cdot)$:

$$\max_{\mathbf{a}} Q_i(s_{it}, \mathbf{a}) = \max_q q d_{it}(q) - \min_{\mathbf{b}} \mathbb{E}_{\hat{\mathbf{q}}_t}\left[\mathbf{g}_{it}(\mathbf{b})^{\text{T}}(\hat{\mathbf{q}}_t - \gamma \mathbf{o}_{it})\right] + \text{const.}, \tag{11}$$

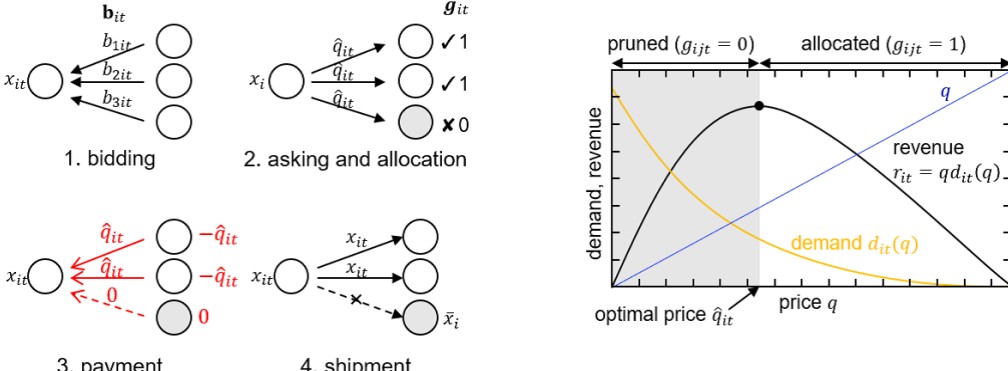

Figure 2: **Left**: Trade process in an envy-free auction. **Right**: Agent price determination curve. Agent revenue is a product of monotonically decreasing demands and prices. The price that maximizes revenue is the optimal price.

where $\mathbf{a} = (\mathbf{b}, q)$. Note that $\mathbf{g}_{it} = H(\mathbf{b} - \mathbf{q}_t)$. The expectation $\mathbb{E}_{\hat{\mathbf{q}}_t}[\cdot]$ was taken because the asking price $\hat{\mathbf{q}}_t$ was unknown for $v_i$, except when $\hat{q}_{it}$ and $g_{iit} = 0$.

Then, to identify the bidding price that $b_{it}$ maximizes returns, the following theorem holds.

**Theorem 3.1.** *(Truthfulness) the optimal bidding price for maximizing returns is* $\hat{\mathbf{b}}_{it} = \gamma \mathbf{o}_{it}$.

This proof is shown in the Appendix A.

This implies agents should only consider their counterfactual returns! When $\gamma = 0$ it is equivalent to a case without auction. Hence, the bidding value is raised if each agent considers their long-time rewards. Consequently, when the NaaA mechanism is used agents behave as if performing valuation for other agents, and declare values truthfully.

Under these conditions, the following corollary holds:

**Corollary 3.1.** *The Nash equilibrium of an envy-free auction is* $(\mathbf{b}_{it}, q_{it})$ *is* $(\gamma \mathbf{o}_{it}, \underset{q}{\arg\max} \, qd_{it}(q))$.

The remaining problem is how to predict $\mathbf{o}_t$. $Q$-learning was used to predict $\mathbf{o}_t$ in this paper as the same way as QUICR Agogino & Tumer (2006). As $\mathbf{o}_{it}$ represented the difference between two $Q$s, each $Q$ was approximated. The state was parameterized using the vector $\mathbf{s}_t$, which contained input and weight. The $\epsilon$-greedy policy with $Q$-learning typically supposed that discrete actions Thus the allocation $g_{ijt}$ was employed as an action rather than $\mathbf{b}_{it}$ and $q_{it}$.

**Algorithm** The overall algorithm is shown in Algorithm 1.

### 3.3 Neuron as an Agent

One benefit of NaaA is that it can be used not only for MARL, but also for network training. Typical neural network training algorithms such as RMSProp (Tieleman & Hinton, 2012) and Adam (Kingma & Ba, 2014) are based on sequential algorithms such as the stochastic gradient descent (SGD). Therefore, the problem they solve can be interpreted as a problem of updating a state (i.e., weight) to a goal (the minimization of the expected likelihood).

### 3.3.1 Adaptive DropConnect

Learning can be accelerated by applying NaaA to the optimizer. In this paper, the application of NaaA to SGD was named *Adaptive DropConnect* (ADC), the finalization of which can be interpreted as a combination of DropConnect (Wan et al., 2013) and Adaptive DropOut (Ba & Frey, 2013). In the subsequent section, ADC is introduced as a potential NaaA application.

---

**Algorithm 1** NaaA: inter-agent reward distribution with envy-free auction

---
1: **for** $t = 1$ **to** $T$ **do**
2:     Compute a bidding price for every edge: **for** $(v_j, v_i) \in \mathcal{E}$ **do** $b_{ijt} \leftarrow \gamma o_{ijt}$
3:     Compute an asking price for every node: **for** $v_i \in \mathcal{V}$ **do** $\hat{q}_{it} \leftarrow \underset{q \in [0,\infty)}{\operatorname{argmax}} q d_{it}(q)$.
4:     **for** $(v_i, v_j) \in \mathcal{E}$ **do**
5:         Compute allocation: $g_{jit} \leftarrow H(b_{jit} - \hat{q}_{it})$
6:         Compute the price the agent should pay: $\rho_{jit} \leftarrow g_{jit} \hat{q}_{it}$
7:     **end for**
8:     Make a payment: **for** $v_i \in \mathcal{V}$ **do** $R_{it} \leftarrow \sum_{j \in N_i^{\text{out}}} \rho_{jit} - \sum_{j \in N_i^{\text{in}}} \rho_{ijt}$,
9:     Make a shipment: **for** $v_i \in \mathcal{V}$ **do** $\tilde{x}_{ijt} = g_{ijt} x_{ijt} + (1 - g_{ijt}) \bar{x}_{ijt}$
10:     **for** $v_i \in \mathcal{V}$ **do**
11:         Observe external state $\mathbf{s}_{it}^{\text{ex}}$
12:         $\mathbf{s}_{it} \leftarrow (\mathbf{s}_{it}^{\text{ex}}, \tilde{\mathbf{x}}_{it}, \boldsymbol{\theta}_i)$, where $\tilde{\mathbf{x}}_{it} = (\tilde{x}_{i1t}, \dots, \tilde{x}_{int})^{\text{T}}$ and $\boldsymbol{\theta}_i$ is $v_i$'s parameter.
13:         Sample action $a_{it}^{\text{ex}} \sim \pi_i^{\text{ex}}(\mathbf{s}_{it})$
14:         Receive external reward $R_{it} \leftarrow R_{it} + R_{it}^{\text{ex}}(a_{it}^{\text{ex}})$
15:         Update $o_{ijt}$ under the manner of $Q$-learning
16:     **end for**
17: **end for**

---

**Algorithm 2** Adaptive DropConnect

---
1: **for** $t = 1$ **to** $T$ **do**
2:     Compute a bidding price for every edge: **for** $(v_j, v_i) \in \mathcal{E}$ **do** $b_{ijt} \leftarrow |w_{ijt}|$
3:     Compute an asking price for every node: **for** $v_i \in \mathcal{V}$ **do** $\hat{q}_{it} \leftarrow \underset{q \in [0,\infty)}{\operatorname{argmax}} q d_{it}(q)$.
4:     **for** $(v_i, v_j) \in \mathcal{E}$ **do**
5:         Compute allocation: $g_{jit} \leftarrow H(b_{jit} - \hat{q}_{it})$
6:     **end for**
7:     Sample a switching matrix $U_t$ from a Bernoulli distribution: $U_t \sim \text{Bernoulli}(\varepsilon)$
8:     Sample the random mask $M_t$ from a Bernoulli distribution: $M_t \sim \text{Bernoulli}(1/2)$
9:     Generate the adaptive mask: $M_t' \leftarrow U_t \circ M_t + (1 - U_t) \circ G_{ijt}$
10:     Compute $\mathbf{h}_t$ for making a shipment: $\mathbf{h}_t \leftarrow (M_t' \circ W_t)\mathbf{x}_t + \mathbf{b}_t$
11:     Update $W_t$ and $\mathbf{b}_t$ by backpropagation.
12: **end for**

---

ADC uses NaaA for supervised optimization problems with multiple revisions. In such problems, the first step is the presentation of an input state (such as an image) by the environment. Agents are expected to update their parameters to maximize the rewards presented by a criterion calculator. Criterion calculators gives batch-likelihoods to agents, representing rewards. Each agent, a classifier, updates its weights to maximize the reward from the criterion calculator. These weights are recorded as an internal state. A heuristic utilizing the absolute value of weight $|w_{ijt}|$ (the technique used by Adaptive DropOut) was applied as the counterfactual return $o_{ijt}$. The absolute value of weights was used because it represented the updated amounts for which the magnitude of error of unit outputs was proportional to $|w_{ijt}|$.

This algorithm is presented as Algorithm 2. Because the algorithm is quite simple, it can be easily implemented and, thus, applied to most general deep learning problems such as image recognition, sound recognition, and even deep reinforcement learning.

## 4 RELATED WORKS

Existing multi-agent reinforcement learning (MARL) communication methods have relied on a trusted third party (TTP) to distribute reward to agents, leaving them inapplicable in peer-to-peer environments. R/DIAL (Foerster et al., 2016) is a communication method for deep reinforcement learning, which train the optimal communication among the agent with Q-learning. It focuses on that paradigm of centralized planning. CommNet (Sukhbaatar et al., 2016), which exploits the characteristics of a unit that is agnostic to the topology of other units, employs backpropagation to train multi-

agent communication. Instead of reward $R(a_t)$ of an agent $i$ for actions at $t$ $a_t$, QUICR-learning (Agogino & Tumer, 2006) maximizes counterfactual reward $R(a_t) - R(a_t - a_{it})$, the difference in the case of the agent $i$ takes an action $a_{it}$ $(a_t)$ and not $(a_t - a_{it})$. COMA (Foerster et al., 2017) also maximizes counterfactual rewards in an actor–critic setting. CommNet, QUICR and COMA have a centralized environment for distributing rewards through a TTP. In contrast, NaaA does not rely on a TTP, and hence, each agent calculates its reward.

While inter-agent reward distribution has not been considered in the context of communication, trading agents have been considered in other contexts. Trading agent competition (TACs), competitions for trading agent design, have been held in various locations regarding topics such as smart grids (Ketter et al., 2013), wholesale (Kuate et al., 2013), and supply chains (Pardoe & Stone, 2009), yielding innumerable trading algorithms such as Tesauro's bidding algorithm (Tesauro & Bredin, 2002) and TacTex'13 (Urieli & Stone, 2014). Since several competitions employed an auction as optimal price determination mechanism (Wellman et al., 2001; Stone et al., 2003), using auctions to determine optimal prices is now a natural approach. Unfortunately, these existing methods cannot be applied to the present situation. First, their agents did not communicate because the typical purpose of a TAC is to create market competition between agents in a zero-sum game. Secondly, the traded goods are not digital goods but instead goods in limited supply, such as power and tariffs. Hence, this is the first paper to introduce inter-agent reward distribution to MARL communications.

Auction theory is discussed in terms of mechanism design (Myerson, 1983), also known as inverse game theory. Second–price auctions (Vickrey, 1961) are auctions including a single product and several buyers. In this paper, a digital goods auction (Guruswami et al., 2005) was used as an auction with an infinite supply. Several methods extend digital goods auction to address collusion, including the consensus estimate (Goldberg & Hartline, 2003) and random sample auction (Goldberg et al., 2006), which can be used to improve our method.

This paper is also related to DropConnect in terms of controlling connections between units. Adaptive DropConnect (ADC), proposed in a later section of this paper as a further application, extends the DropConnect (Wan et al., 2013) regularization technique. The finalized idea of ADC (which uses a skew probability correlated to the absolute value of weights rather than dropping each connection between units by a constant probability) is closer to Adaptive DropOut (Ba & Frey, 2013), although their derivation differs. The adjective "adaptive" is added with respect to the method. Neural network optimizing using RL was investigated by Andrychowicz et al. (2016); however, their methods used a recurrent neural network (RNN) and are therefore difficult to implement, whereas the proposed method is RNN-free and forms as a layer. For these reasons, its implementation is simple and fast and it also has a wide area of applicability.

## 5 EXPERIMENT

To confirm that NaaA works widely with machine learning tasks, we confirm our method of supervised learning tasks as well as reinforcement learning tasks. As supervised learning tasks, we use typical machine learning tasks such as image classification using MNIST, CIFAR-10, and SVHN.

As reinforcement tasks, we confirm single- and multi-agent environment. The single-agent environment is from OpenAI Gym. We confirm the result using a simple reinforcement task: CartPole. In multi-agent, we use ViZDoom, a 3D environment for reinforcement learning.

### 5.1 CLASSIFICATION

#### 5.1.1 SETUP

For classification, three types of datasets were used: MNIST, CIFAR-10, and STL-10. The given task was to predict the label of each image, and each dataset had a class number of 10. The first dataset, MNIST, was a collection of black and white images of handwritten digits sized $28 \times 28$. The training and test sets contained 60,000 and 10,000 example images, respectively. The CIFAR-10 dataset images were colored and sized $32 \times 32$, and the assigned task was to predict what was shown in each picture. This dataset contained 6,000 images per class (5,000 for training and 1,000 for testing). The STL-10 dataset was used for image recognition, and had 1,300 images for each class (500 training, 800 testing). Each image was sized $96 \times 96$; however, for the experiment, the

images were resized to 48 × 48 because the greater resolution of this dataset (relative to the above datasets) required far more computing time and resources.

### 5.1.2 MODEL

Two models were compared in this experiment: DropConnect and Adaptive DropConnect (the model proposed in this paper). The baseline model was composed of two convolutional layers and two fully connected layers whose outputs are dropped out (we set the possibility as 0.5). The labels of input data were predicted using log-softmaxed values from the last fully connected layer. In the DropConnect and Adaptive DropConnect models, the first fully connected layer was replaced by a DropConnected and Adaptive DropConnected layer, respectively. It should be noted that the DropConnect model corresponded to the proposed method when $\varepsilon = 1.0$, meaning agents did not perform their auctions but instead randomly masked their weights.

### 5.1.3 RESULTS

The models were trained over ten epochs using the MNIST datasets, and were then evaluated using the test data. The CIFAR-10 and STL-10 epoch numbers were 20 and 40, respectively. Experiments were repeated 20 times for each condition, and the averages and standard deviations of error rates were calculated. Results are shown in Table 1. As expected, the Adaptive DropConnect model performed with a lower classification error rate than either the baseline or DropConnect models regardless of the given experimental datasets.

Table 1: Experimental result for image classification tasks and single-agent RL

|  | MNIST | CIFAR-10 | STL-10 | CartPole |
|---|---|---|---|---|
| DropConnect (Wan et al., 2013) | $1.72 \pm 0.160$ | $43.14 \pm 1.335$ | $50.92 \pm 1.322$ | $285 \pm 21.5$ |
| Adaptive DropConnect | $\mathbf{1.36} \pm 0.132$ | $\mathbf{39.84} \pm 1.035$ | $\mathbf{42.17} \pm 2.329$ | $\mathbf{347} \pm 29.4$ |

### 5.2 SINGLE-AGENT RL

Next, the single-agent reinforcement learning task was set as the CartPole task from OpenAI Gym (Brockman et al., 2016) with visual inputs. In this setting, the agent was required to balance a pole while moving a cart. The images contained a large amount of non-useful information, making pixel pruning important. The result in Table 1 demonstrates that our method improves the standard RL.

### 5.3 MULTI-AGENT RL

The proposed reward distribution method was confirmed to work as expected by a validation experiment using the multi-agent setting in ViZDoom (Kempka et al., 2016), an emulator of Doom containing a map editor where additional agents complement the main player. A main player in the ViZDoom environment aims to seek the enemy in the map and then defeat the enemy.

### 5.3.1 SETUP

A defend the center (DtC)-based scenario, provided by ViZDoom platform, was used for this experiment. Two players, a main player and a cameraman, were placed in the DtC, where they started in the center of a circular field and then attacked enemies that came from the surrounding wall. Although the main player could attack the enemy with bullets, the cameraman had no way to attack, only scouting for the enemy. The action space for the main player was the combination of { attack, turn left, turn right }, giving a total number of actions $2^3 = 8$. The cameraman had two possible actions: { turn left, turn right }. Although the players could change direction, they could not move on the field. Enemies died after receiving one attack (bullet) from the main player, and then player received a score of +1 for each successful attack. The main player received 26 bullets by default at the beginning of each episode. The main player died if they received attacks from the enemy to the extent that their health dropped to 0, and received a score of -1 for each death. The cameraman did not die if attacked by an enemy. Episodes terminated either when the maim player died or after 525 steps elapsed.

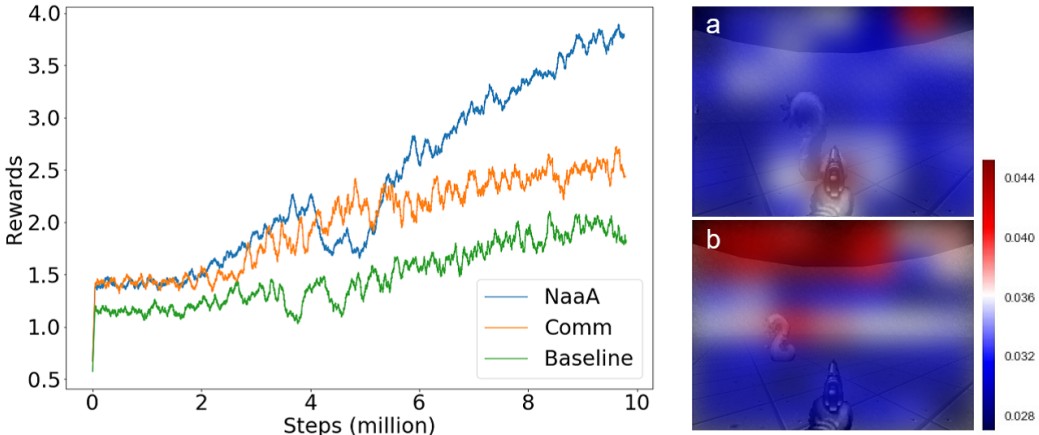

Figure 3: **Left:** Learning curve for ViZDoom multi-agent task. The proposed NaaA–based method outperformed the other two methods (baseline and Comm DQNs). **Right:** Visualizing reward from the main player to the cameramann shows us what is important information for the main player: (a) The pistol. (b) The point at which the enemy appeared and approached.

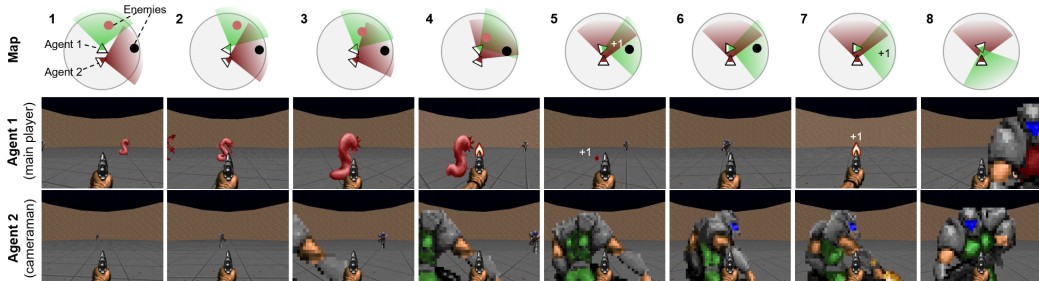

Figure 4: NaaA leads agents to enter a cooperative relationship. First, the two agents face different directions, and the cameraman sells their information to the main player (**1**). The main player (information buyer) starts to turn right to find the enemy. The cameraman (information seller) starts to turn left to seek new information by finding the blind area of the main player (**2** and **3**). After turning, the main player attacks the first, having already identified enemy (**4** and **5**). Once the main player finds the enemy, he attacks and obtains the reward (**6** and **7**). Both agents then return to watching the dead area of the other until the next enemy appears (**8**).

### 5.3.2 MODEL

Three models, described below, were compared: the proposed method and two comparison targets.

*Baseline*: DQN without communication. The main player learned standard DQN with the perspective that the player is viewing. Because the cameraman did not learn, this player continued to move randomly.

*Comm*: DQN with communication, inspired by Commnet. The main player learns DQN with two perspectives: theirs and that of the cameraman. The communication vector is learned with a feed-forward neural network.

*NaaA*: The proposed method. The main player learned DQN with two perspectives: theirs and that of the cameraman. Transmissions of rewards and communications were performed using the proposed method.

### 5.3.3 RESULTS

Training was performed over the course of 10 million steps. Figure 3 Left demonstrates the proposed NaaA model outperformed the other two methods. Improvement was achieved by Adaptive DropConnect. It was confirmed that the cameraman observed the enemy through an episode, which could be interpreted as the cameraman reporting enemy positions. In addition to seeing the enemy, the cameraman observed the area behind the main player several times. This enabled the cameraman to observe enemy attacks while taking a better relative position.

To further interpret this result, a heatmap visualization of revenue earned by the agent is presented in Figure 3 Right. The background picture is a screen from Doom, recorded at the moment when the CNN filter was most activated. Figure 4 shows an example of learnt sequence of actions by our method.

## 6 CONCLUDING REMARKS AND FUTURE WORK

This paper proposed a NaaA model to address communication in MARL without a TTP based on two key ideas: inter-agent reward distribution and auction theory. Existing MARL communication methods have assumed the existence of a TTP, and hence could not be applied in peer–to–peer environments. The inter-agent reward distribution, making agents redistribute the rewards they received from the internal/external environment, was reviewed first. When an envy-free auction was introduced using auction theory, it was shown that agents would evaluate the counterfactual returns of other agents. The experimental results demonstrated that NaaA outperformed a baseline method and a CommNet-based method.

Furthermore, a $Q$-learning based algorithm, termed Adaptive DropConnect, was proposed to dynamically optimize neural network topology with counterfactual return evaluation as a further application. To evaluate this application, experiments were performed based on a single-agent platform, demonstrating that the proposed method produced improved experimental results relative to existing methods.

Future research may also be directed toward considering the connection between NaaA and neuroscience or neuroevolution. Edeleman propounded the concept of neural Darwinism (Edelman, 1987), in which group selection occurs in the brain. Inter-agent rewards, which were assumed in this paper, correspond to NTFs and could be used as a fitness function in genetic algorithms for neuroevolution such as hyperparameter tuning.

As NaaA can be applied in peer-to-peer environments, the implementation of NaaA in blockchain (Swan, 2015) is under consideration. This implementation would extend the areas where deep reinforcement learning could be applied. Bitcoin (Nakamoto, 2008) could be used for inter-agent reward distribution, and the auction mechanism could be implemented by smart contracts (Buterin et al., 2014). Using the NaaA reward design, it is hoped that the world may be united, allowing people to share their own representations on a global scale.

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

## APPENDIX

### A.1 PROOF OF THEOREM 3.1

The optimization problem in Eq:11 is made of two terms except of the constant, and the only second term is depends on $\mathbf{b}$. Hence, we consider to optimize the second term. The optimal bidding prices $\hat{\mathbf{q}}_t$ is given by the following equation.

$$
\hat{\mathbf{b}}_{it} = \operatorname*{argmin}_{\mathbf{b}} \mathbb{E}_{\mathbf{q}_t} \left[ \mathbf{g}_{it}(\mathbf{b})^{\mathrm{T}}(\mathbf{q}_t - \gamma \mathbf{o}_{it}) \right] = \operatorname*{argmin}_{\mathbf{b}} \mathbb{E}_{\mathbf{q}_t} \left[ H(\mathbf{b} - \mathbf{q}_t)^{\mathrm{T}}(\mathbf{q}_t - \gamma \mathbf{o}_{it}) \right]
$$

$$
= \operatorname*{argmin}_{\mathbf{b}} \mathbb{E}_{\mathbf{q}_t} \left[ \sum_{j=1}^{N} H(b_j - q_{jt})(q_{jt} - \gamma o_{ijt}) \right] = \operatorname*{argmin}_{\mathbf{b}} \sum_{j=1}^{N} \mathbb{E}_{q_{jt}} \left[ H(b_j - q_{jt})(q_{jt} - \gamma o_{ijt}) \right],
\tag{12}
$$

From independence, the equation is solved if we solve the following problem.

$$
\hat{b}_{ijt} = \operatorname*{argmin}_{b} \mathbb{E}_{q_{jt}} \left[ H(b - q_{jt})(q_{jt} - \gamma o_{ijt}) \right], \ \ \forall j \in \{1, \dots, N\}
\tag{13}
$$

Hence, $\hat{b}_{ijt}$ can be derived as the solution which satisfies the following equation.

$$
\frac{\partial}{\partial b} \mathbb{E}_{q_{jt}} \left[ H(b - q_{jt})(q_{jt} - \gamma o_{ijt}) \right] \bigg|_{b=\hat{b}_{ijt}} = 0, \quad \frac{\partial^2}{\partial b^2} \mathbb{E}_{q_{jt}} \left[ H(b - q_{jt})(q_{jt} - \gamma o_{ijt}) \right] \bigg|_{b=\hat{b}_{ijt}} > 0
$$

For simplicity, we let $q = q_{jt}$ and $o = o_{ij,t+1}$. Then, the following equation holds.

$$
\frac{\partial}{\partial b} \mathbb{E}_q \left[ H(b - q)(q - \gamma o) \right] = \frac{\partial}{\partial b} \int_0^{\infty} H(b - q)(q - \gamma o) p(q) dq
$$

$$
= \frac{\partial}{\partial b} \int_0^{b} (q - \gamma o) p(q) dq
$$

$$
= (b - \gamma o) p(q = b),
\tag{14}
$$

$$
\frac{\partial^2}{\partial b^2} \mathbb{E}_q \left[ H(b - q)(q - \gamma o) \right] = p(q = b) + (b - \gamma o) \frac{\partial}{\partial b} p(q = b)
\tag{15}
$$

Then, condition $\frac{\partial}{\partial b} \mathbb{E}_{q_{jt}} \left[ H(b - q_{jt})(q_{jt} - \gamma o_{ijt}) \right] |_{b=\hat{b}_{ijt}} = 0$ is satisfied only by $\hat{b}_{ijt} = \gamma o_{ijt}$. We substitue $\hat{b}_{ijt}$ into Eq:15,

$$\frac{\partial^2}{\partial b^2} \mathbb{E}_{q_{jt}} \left[ H(b - q_{jt})(q_{jt} - \gamma o_{ijt}) \right] \bigg|_{b=\hat{b}_{ijt}} = p(q = \hat{b}_{ijt}) + 0 > 0 \tag{16}$$

Therefore, $\hat{b}_{ijt} = \gamma o_{ijt}$ is a unique solution as the minimum point. From generality, $\hat{\mathbf{b}}_{it} = \gamma \mathbf{o}_{it}$ holds.

