# OpenReview forum: "Neuron as an Agent"
_ICLR.cc/2018/Conference — Invite to Workshop Track_

### Official Review · AnonReviewer1 · 2017-11-27
**Good work. Presentation needs further polishing.**

**Rating:** 7
**Confidence:** 3

**Review:**

In this paper, the authors present a novel way to look at a neural network such that each neuron (node) in the network is an agent working to optimize its reward. The paper shows that by appropriately defining the neuron level reward function, the model can learn a better policy in different tasks. For example, if a classification task is formulated as reinforcement learning where the ultimate reward depends on the batch likelihood, the presented formulation (called Adaptive DropConnect in this context) does better on standard datasets when compared with a strong baseline.

The idea proposed in the paper is quite interesting, but the presentation is severely lacking. In a work that relies heavily on precise mathematical formulation, there are several instances when the details are not addressed leading to ample confusion making it hard to fully comprehend how the idea works. For example, in section 5.1, notations are presented and defined much later or not at all (g_{jit} and d_{it}). Many equations were unclear to me for similar reasons to the point I decided to only skim those parts. Even the definition of external vs. internal environment (section 4) was unclear which is used a few times later. Like, what does it mean when we say, “environment that the multi-agent system itself touches”?

Overall, I think the idea presented in the paper has merit, but without a thorough rewriting of the mathematical sections, it is difficult to fully comprehend its potential and applications.

---

> ### Author Response · Authors · 2017-12-03
> **All right. We will improve the mathematical formulation**
>
> Thank you for reading and commenting our paper.  We will polish our mathematical formulation to improve your understanding during this period.
>
> > Even the definition of external vs. internal environment (section 4) was unclear which is used a few times later.
>
> An external environment is the original environment such as Doom and Atari, and an internal environment is a set of units. From an agent's perspective, other units are considered as an environment.
>
> The quick reference can also be helpful.
>
>                    Environment for a unit         State for a unit                               Observation for a unit
>                    -----------------------------------   -------------------------------------------   ---------------------------------------
> External    original environment            original state                                   original observation
> Internal     other units                              activation of all the other units   activation of allocated units
> Both           -                                                -                                                         be used to predict o_{ijt}
>
>                    Reward per unit                     Total reward over units
>                    -----------------------------------   -------------------------------------------------------------
> External    original reward                       total original reward (designer's objective)
> Internal     revenue from units - cost     0
> Both           units' objective                       total original reward (designer's objective)
>
>
> > “environment that the multi-agent system itself touches”
>
> Typically, there is boundary between an agent and an environment (e.g., a robot in a room). We wrote this situation that the agent with a NN (as the multi-agent system) touches the environment.
>
> > In section 5.1, notations are presented and defined much later or not at all (g_{jit} and d_{it}).
> > Many equations were unclear to me for similar reasons to the point
> > Without a thorough rewriting of the mathematical sections, it is difficult to fully comprehend its potential and applications.
>
> As we will reflect the comments to our paper, please wait for it.

---

### Official Review · AnonReviewer2 · 2017-11-28
**Requires major editing to be fit for publication.**

**Rating:** 3
**Confidence:** 5

**Review:**

The authors consider a Neural Network where the neurons are treated as rational agents. In this model, the neurons must pay to observe the activation of neurons upstream. Thus, each individual neuron seeks to maximize the sum of payments it receives from other neurons minus the cost for observing the activations of other neurons (plus an external reward for success at the task).

While this is an interesting idea on its surface, the paper suffers from many problems in clarity, motivation, and technical presentation. It would require very major editing to be fit for publication.

The major problem with this paper is its clarity. See detailed comments below for problems just in the introduction. More generally, the paper is riddled with non sequiturs. The related work section mentions Generative Adversarial Nets. As far as I can tell, this paper has nothing to do with GANs. The Background section introduces notation for POMDPs, never to be used again in the entirety of the paper, before launching into a paragraph about apoptosis in glial cells.

There is also a general lack of attention to detail. For example, the entire network receives an external reward (R_t^{ex}), presumably for its performance on some task. This reward is dispersed to the the individual agents who receive individual external rewards (R_{it}^{ex}). It is never explained how this reward is allocated even in the authors’ own experiments. The authors state that all units playing NOOP is an equilibrium. While this is certainly believable/expected, such a result would depend on the external rewards R_{it}^{ex}, the observation costs \sigma_{jit}, and the network topology. None of this is discussed. The authors discuss Pareto optimality without ever formally describing what multi-objective function defines this supposed Pareto boundary. This is pervasive throughout the paper, and is detrimental to the reader’s understanding.

While this might be lost because of the clarity problems described above, the model itself is also never really motivated. Why is this an interesting problem? There are many ways to create rational incentives for neurons in a neural net. Why is paying to observe activations the one chosen here? The neuroscientific motivation is not very convincing to me, considering that ultimately these neurons have to hold an auction. Is there an economic motivation? Is it just a different way to train a NN?

Detailed Comments:
“In the of NaaA” => remove “of”?
“passing its activation to the unit as cost” => Unclear. What does this mean?
“performance decreases if we naively consider units as agents” => Performance on what?
“.. we demonstrate that the agent obeys to maximize its counterfactual return as the Nash Equilibrium“ => Perhaps, this should be rewritten as “Agents maximize their counterfactual return in equilibrium.
“Subsequently, we present that learning counterfactual return leads the model to learning optimal topology” => Do you mean  “maximizing” instead of learning. Optimal with respect to what task?
“pure-randomly” => “randomly”
 “with adaptive algorithm” => “with an adaptive algorithm”
“the connection” => “connections”
“In game theory, the outcome maximizing overall reward is named Pareto optimality.” => This is simply incorrect.

---

> ### Author Response · Authors · 2017-12-03
> **(cont'd) Response**
>
> Response to Paragraph 5:
>
> > There are many ways to create rational incentives for neurons in a neural net.
>
> As I don’t think there are many methods for our problem setting, please provide a link.
>
> > The neuroscientific motivation is not very convincing to me, considering that ultimately these neurons have to hold an auction.
>
> Auction is more than auction as it used in mechanism design to orchestrate the actions of agents with mechanism. So, think out of the box, and throw away the typical image of auction.
>
> > Is there an economic motivation? Is it just a different way to train a NN?
>
> Yes, there is an economic motivation as well as to improve training a NN.
>
>
> Response to Paragraph 6 (the detailed comments):
>
> > “passing its activation to the unit as cost” => Unclear. What does this mean?
>
> "to observe their activation" is correct.
> (As it was a mistake in the native check process, we will change the native checker later)
>
> > “performance decreases if we naively consider units as agents” => Performance on what?
>
> Performance on the total cumulative external reward.
>
> > “Subsequently, we present that learning counterfactual return leads the model to learning optimal topology” => Do you mean  “maximizing” instead of learning.  Optimal with respect to what task?
>
> Just like the above answer, it will be optimal with respect to the total cumulative external reward.

---

> ### Author Response · Authors · 2017-12-03
> **A month is enough to meet your requirements**
>
> Thank you for reading and commenting our paper.  Let us answer the question first:
>
> > The model itself is also never really motivated. Why is this an interesting problem?
>
> Do you know “Blockchain”, the technology supporting most of virtual currencies such as Bitcoin and Ethereum, which has vast market cap of $100 billions? Also, there was a news that the price of one Bitcoin exceeded $10,000 last week though the price was $1,000 at the beginning of this year. The emerging technology enable us to send incentive among agents in a decentralized environment. If an agent can earn money with realistic way such as automatic financial trading for stock, debt and coins, the way agent takes will be reinforcement learning, and it would face a problem of POMDP, because the market is inefficient in which an agent who has  informative data can gain advantage. Hence, the agent will buy the informative data from other agent by paying incentive over the multi-agent setting, and the incentive will be distributed in the Blockchain environment.
>
> Such background raises the question in a face of the paper:
>      “will reinforcement learning work even if we consider each unit as an autonomous agent?”
> which motivates our framework. To answer the question, there are several issues to address such as “how much is appropriate reward the agent should pay?” and “how to address the social dilemma?”. All the answers are written in the paper.
>
> If the major problem is clarity as you mentioned, a month will be enough to solve it.
>
>
> Response to Paragraph 3:
>
> > GANs
>
> Although our paper had nothing to do with GANs directly, we mentioned GAN as a game-theoretic approach to model the real environment.
>
> > POMDPs, never to be used again in the entirety of the paper
>
> Notation of POMDP is used in methods such as S_O (below Eq (3)) and \gamma (in Eq (4)).
> Besides, Eq (1) in the section of PODMP is used to derive Eq (9).
>
>
> Response to Paragraph 4:
>
> > It is never explained how this reward is allocated even in the authors’ own experiments.
>
> In the classification and the single-agent setting, the reward is given only to the endpoint of agent.
>
> In the multi-agent setting,  the external reward (reward from the Doom environment) is given to the agents (a main player and a cameraman) with following ways.
>     1. Baseline: endpoint of the main player.
>     2. Comm: endpoint of the main player and the cameraman (the same configuration to the original paper of CommNet)
>     3. NaaA: endpoint of the main player. The reward is pour from the main player to the cameraman as an internal reward.
>
> > The authors state that all units playing NOOP is an equilibrium. While this is certainly believable/expected, such a result would depend on the external rewards R_{it}^{ex}, the observation costs \sigma_{jit}, and the network topology. None of this is discussed.
>
> Although the few agents which can gain the external reward can survive, most of the agents whose R_{it}^{ex} equals to 0 becomes NOOP regardless of its network topology because \sigma_{jit} will equal 0 at the convergence. As we will post the revised version which contains the proof, please wait for it.
>
> > The authors discuss Pareto optimality without ever formally describing what multi-objective function defines this supposed Pareto boundary. This is pervasive throughout the paper, and is detrimental to the reader’s understanding.
>
> The objective function is return (discounted cumulative reward). That is,
>                   Σ_{t=0}^T [ γ^t R_t^{ex} ],
> where R_{ex,t} := Σ_i R_{it}^{ex} is overall reward from the external environment. Pareto optimal is defined for the objective functions of all the agents.
>
> (continues...)

---

### Official Review · AnonReviewer4 · 2017-12-18
**Interesting and promising ideas, but need significant polishing and substantiation**

**Rating:** 6
**Confidence:** 4

**Review:**

This paper proposed a novel framework Neuron as an Agent (NaaA) for training neural networks to perform various machine learning tasks, including classification (supervised learning) and sequential decision making (reinforcement learning). The NaaA framework is based on the idea of treating all neural network units as self-interested agents and optimizes the neural network as a multi-agent RL problem. This paper also proposes adaptive dropconnect, which extends dropconnect (Wan et al., 2013) by using an adaptive algorithm for masking network topology.

This work attempts to bring several fundamental principles in game theory to solve neural network optimization problems in deep learning. Although the ideas are interest and technically sound, and the proposed algorithms are demonstrated to outperform several baselines in various machine learning tasks, there several major problems with this paper, including lacking clarity of presentation, insights and substantiations of many claims. These issues may need a significant amount of effort to fix as I will elaborate more below.

1. Introduction
There are several important concepts, such as reward distribution, credit assignment, which are used (from the very beginning of the paper) without explanation until the final part of the paper.

The motivation of the work is not very clear. There seems to be a gap between the first paragraph and the second paragraph. The authors mentioned that “From a micro perspective, the abstraction capability of each unit contribute to the return of the entire system. Therefore, we address the following questions. Will reinforcement learning work even if we consider each unit as an autonomous agent ”
Is there any citation for the claim “From a micro perspective, the abstraction capability of each unit contribute to the return of the entire system” ?  It seems to me this is a very general claim. Even RL methods with linear function approximations use abstractions.  Also, it is unclear to me why this is an interest question. Does it have anything to do with existing issues in DRL? Moreover, The definition of autonomous agent is not clear, do you mean learning agent or policy execution agent?

“it uses \epsilon-greedy as a policy, …” Do you mean exploration policy?
I also have some concerns regarding the claim that “We confirm that optimization with the framework of NaaA leads to better performance of RL”. Since there are only two baselines are compared to the proposed method, this claim seems too general to be true.

It is not clear to why the authors mention that “negative result that the return decreases if we naively consider units as agents”. What is the big picture behind this claim?

“The counterfactual return is that by which we extend reward …” need to be rewritten.

The last paragraph of introduction discussed the possible applications of the proposed methods without any substantiation, especially neither citations nor any related experiments of the authors are provided.

2 Related Work

“POSG, a class of reinforcement learning with multiple ..” -> reinforcement learning framework

“Another one is credit assignment. Instead of reward.. ” Two sentences are disconnected and need to be rewritten.

“This paper unifies both issues” sounds very weird. Do you mean “solves/considers both issues in a principled way”?

The introduction of GAN is very abrupt. Rather than starting from introducing those new concepts directly, it might be better to mention that the proposed method is related to many important concepts in game theory and GANs.

“,which we propose in a later part of this paper” -> which we propose in this paper


3. Background

“a function from the state and the action of an agent to the real value” -> a reward function

Should provide a citation for DRQN

There is a big gap between the last two paragraphs of section 3.

4. Neuro as an agent

“We add the following assumption for characteristics of the v_i” -> assumptions for characterizing v_i

“to maximize toward maximizing its own return” -> to maximize its own return

We construct the framework of NaaA from the assumptions -> from these assumptions

“indicates that the unit gives additional value to the obtained data. …” I am not sure what this sentence means, given that \rho_ijt is not clearly defined.

5. Optimization

“NaaA assumes that all agents are not cooperative but selfish” Why? Is there any justification for such a claim?


What is the relation between \rho_jit and q_it ?

“A buyer which cannot receive the activation approximates x_i with …” It is unclear why a buyer need to do so given that it cannot receive the activation anyway.

“Q_it maximizing the equation is designated as the optimal price.” Which equation?

e_j and 0 are not defined in equation 8


6 Experiment
setare -> set are

what is the std for CartPole in table 1

It is hard to judge the significance of the results on the left side of figure 2. It might be better to add errorbars to those curves

More description should be provided to explain the reward visualization on the right side of figure 2. What reward? External/internal?

“Specifically, it is applicable to various methods as described below …” Related papers should be cited.

---

> ### Author Response · Authors · 2017-12-31
> **Helpful comments. Thank you so much**
>
> Thank you for reading and commenting our paper.
> We really appreciate your detailed comments. Most of them were very helpful to brush up our paper.
> We are about to finalize the paper, and will upload a version which highly improved clarity at 5th Jan.
> So, please look forward it.
>
> Enjoy the holidays & Have a happy new year.

---

### Public Comment · ~Xin_Yang1 · 2017-11-11
**Interesting paper. I have three questions**

Very interesting paper. It shows a novel framework to consider all the units as agents.
Even though the problem setting is challenging, the paper solved it by converting it into a scheme of counterfactual return maximization using an elegant trick from auction theory.

Nonetheless, I have several questions about the paper.
1. What is the objective function?  While the author states the problem is POSG, I guess the problem is POMDP/MDP since the paper introduced a Doom-based environment as the experiment. I'm not sure to what the algorithm want to maximize actually.
2. I'm unsure how to predict o_it actually. Though it seems to use Q-learn according to the paper, I want you to provide detail information of the architecture.
3. As I guess ViZDoom is a single-agent platform, how do you realize the multi-agent setting?  I mean, are there some special implementations?

There are minor comments which may improve your paper:
 - Definition of R and \pi is missing. I supposed they are a reward function and a policy.
 - Provide definition of (s_it^ex, \tilde x, \theta_i) in line 12, algo 1.

---

> ### Author Response · Authors · 2017-11-11
> **Thank you for being interested in our work**
>
>
> 1) The objective function is return (discounted cumulative reward). That is,
>                   Σ_{t=0}^T [ γ^t R_t^{ex} ],
>     where R_{ex,t} := Σ_i R_{it}^{ex} is overall reward from the external environment.
>
>    > POMDP/MDP
>    The actual problem we want to solve is POMDP.
>    However, we extended it to POSG, multi-agent problem, because we consider bunch of neurons as agents.
>    That's why we distinguished it external/internal environment in the paper.
>
> 2) Yes, we used DQN-like architecture (Q-learning with neural net and experience replay) to predict counterfactual return of j for i  at t o_{ijt}. The detail is as below.
>      - The input is a state s_{it}, a coupled vector made of an external state, input vector, and parameter (weight and bias).
>      - The output is Q-value Q(s_{it}, g_{ijt}), where g_{ijt} \in {0, 1} is allocation. Hence, there are 2 |N_i^{in}| Q-values per unit, where |N_i^{in}| is number of j's (indices of connected units from a unit v_i).
>      - o_{ijt} is calculated from a pair of scalars from the output: Q(s_{it}, 1) - Q(s_{it}, 0).
>      - The model made of one layer, but also deeper architectures can be introduced.
>
> 3) ViZDoom partially supports multi-agent environment, but it does not supports communication among the agents.
>     So, we extended it with writing the original code which supports communication.
>
> > def of (s_it^ex, \tilde{x}, \theta_i)
>
> The coupled vectors are designed as a state to predict Q-values for o_{ijt}.
> Here is the definition of the each notation.
> - s_it^ex: external state
> - \tilde{x}: the predicted input vector from limited information. \tilde{x} := x * g + \bar{x} * (1-g).
>    \bar{x} is mean value of x.
> - \theta_i: the parameter of v_i. For example, weight and bias for linear unit.
> Please also see our answer (2) in this post.

---

### Public Comment · (anonymous) · 2017-11-18
**External/internal**

Why do you just divide the environment into two types: external/internal?
There is also another way to simply use neural network.

---

> ### Author Response · Authors · 2017-11-19
> **Re: External/internal**
>
> Good questions.
>
> > External/internal
>
> In reinforcement learning (RL), there are two parts: an environment and an agent.
> In "deep" RL, there is a neural network inside the agent as a value/policy function approximator.
> The network contains bunch of units,
> and NaaA considers the network as a multi-agent system, and each unit as an agent.
> From perspective of the unit, the other units are considered as an environment.
> To distinguish from the original environment, we call it an internal environment,
> and call the original environment an external environment.
>
> Here is a quick reference which can also be helpful.
>
>                    Environment for a unit         State for a unit                               Observation for a unit
>                    -----------------------------------   -------------------------------------------   ---------------------------------------
> External    original environment            original state                                   original observation
> Internal     other units                              activation of all the other units   activation of allocated units
> Both           -                                                -                                                         be used to predict o_{ijt}
>
>                    Reward per unit                     Total reward over units
>                    -----------------------------------   -------------------------------------------------------------
> External    original reward                       total original reward (designer's objective)
> Internal     revenue from units - cost     0
> Both           units' objective                       total original reward (designer's objective)
>
>
> > Why not use simple neural network
>
> Suppose AIs had ego. That is, they maximize not total reward but their own reward in a multi-agent system.
> Although recent works such as CommNet supposed cooperate setting, say, all the agents have obtain total reward R,
> if the agents were selfish, there would be no incentive to cooperate, and hence they would not communicate each other.
> The problem is known as social dilemma (e.g., prisoner's dilemma), and leads the overall reward.
> NaaA enables us to design such a multi-agent setting.
> Also, NaaA can be used multi-agent setting in which the agents are made by other people.

---

### Public Comment · ~Luis_Garcia1 · 2017-11-19
**Cool work!**

The method is general and that makes it widely applicable for many problems.
I hope that the design concept will be a new basis of studies such as GAN.

I found very challenging trying to find alternative AI patterns or routines
based on the cooperation of two AIs. I would like to see this happening more often in videogames.
 Are there any demo videos?

However, your idea, Adaptive Dropconnect seems to be complicated to Implement.
How can we implement it?

---

> ### Author Response · Authors · 2017-11-19
> **Demo and sample code**
>
> Thank you for being interested in.
>
> > Demo
>
> You can see our demo for the multi-agent Doom in the following URL.
> https://youtu.be/paT2n40QHOA
>
> > Implementation of Adaptive Dropconnect
>
> Implementation is easy because you can use it by just replace a layer.
>
> Here is a sample vanilla code w/o Adaptive DropConnect in pytorch:
>
>  1  class Net(nn.Module):
>  2     def __init__(self):
>  3         super(Net, self).__init__()
>  4         self.conv1 = nn.Conv2d(3, 10, kernel_size=5)
>  5         self.conv2 = nn.Conv2d(10, 20, kernel_size=5)
>  6         self.fc1 = nn.Linear(500,100)
>  7         self.fc2 = nn.Linear(100, 10)
>  8
>  9     def forward(self, x):
>  10         x = F.relu(F.max_pool2d(self.conv1(x), 2))
>  11         x = F.relu(F.max_pool2d(self.conv2(x), 2))
>  12         x = x.view(-1, 500)
>  13         x = F.relu(self.fc1(x, training=self.training))
>  14         x = F.dropout(x, training=self.training)
>  15         x = self.fc2(x)
>  16         return F.log_softmax(x)
>
> You can turn on Adaptive DropConnnect by just replace a line with
>   6         self.fc1 = nn.Linear(500,100)
>                                   vvv
>   6         self.fc1 = TradeLinear(500,100,eps=0.2)
> TradeLinear is contained in our provided library, which supports Adaptive DropConnnect and NaaA.

---

### Author Response · Authors · 2018-01-05
**The New Revision**

We uploaded the revised version of our paper.

As you can see, over 80% of the paper is major edited to improve clarity while the claim is same as the previous version.

Especially, we make it clear the motivation and the method.
Please read throughout the paper again.

---

### Public Comment · ~Kevin_James_Baker1 · 2018-03-02
**Could it actually work that way in biology: "Neuron as an Agent"?**

I haven't gone through the paper to examine all the formulas, nor reproduced the results.  I am interested in the neuroscience connection, and if the paper is still being worked on perhaps that could be elaborated upon.  Is it suggested that biological neurons might act as agents?﻿  If so, then what is the biological environment for a biological neuron acting as an agent and what are the biological rewards/punishment?

You have opened an interesting can of worms.  I'd like to write a paper that would read like this in abstract:
<sketch of abstract. No title yet>
Neurons move during fetal development to their final location.  Glial cells provide guidance to the neurons- hence acting as their environment.
https://www.ncbi.nlm.nih.gov/books/NBK10831/

Why should the glial cells stop acting as an environment for the neurons after that.  They provide structural support and nourishment; for example, astrocyte glial cells even have synaptic connections with neurons and modulate the synaptic connections between neurons.
https://www.ncbi.nlm.nih.gov/pmc/articles/PMC4527946/

I seems safe to say glial cells provide some of the rewards/punishments, whereas the other neurons that are connected also provide rewards and punishments and act like the deep Q.
My paper would go on to suggest a hypothesis that consciousness is a continuum and is distributed holographically over all neurons and glia in the brain, and by acting as an agent each neuron is exhibiting a small piece of consciousness.  It is aware like an amoeba choosing to go one direction or another that it is exploring its environment of glia cells and other neurons.
https://arxiv.org/abs/0810.4179
http://www.ingentaconnect.com/content/imp/jcs/2004/00000011/00000001/1400

I would go on to address this complaint in this NPR link below, by citing your paper to show how we can emulate complex heterogeneous neurons that embody a small amount of conciousness in AGI:
https://www.npr.org/sections/13.7/2014/11/21/365753466/artificial-intelligence-really-is-pseudo-intelligence
<end abstract>

I have to see if everything I suggest has already been published by reading up on Neural Darwinism:
https://scholar.google.com/scholar?hl=en&as_sdt=0%2C38&as_ylo=2014&as_vis=1&q=neural+darwinism+and+consciousness&btnG=&oq=Neural+Darw

---

### Decision · Program_Chairs · 2018-01-29
**ICLR 2018 Conference Acceptance Decision**

**Decision:**

Invite to Workshop Track

**Comment:**


The reviewers have significantly different views, with one strongly negative,
one strongly positive, and one borderline negative.  However, all three
reviews seem to regard the NaaA framework as a very interesting and novel approach to training neural nets.  They also concur that the major issue with the paper is very confusing technical exposition regarding the motivation, math details, and how the idea works.  The authors indicate that they have significantly revised the manuscript to improve the exposition, but none of the reviewers have changed their scores.  One reviewer states that "technical details are still too heavy to easily follow."  My own take regarding the current section 3 is that it is still very challenging to parse and follow. Given this analysis, the committee recommends this for workshop.

Pros:
        Interesting and novel framework for training NNs
        "Adaptive DropConnect" algorithm contribution
        Good empirical results in image recognition and ViZDoom domains

Cons:
        Technical exposition is very challenging to parse and follow
        Some author rebuttals do not inspire confidence.  For example,
motivation of method due to "$100 billion market cap of Bitcoin" and in reply to unconvincing neuroscience motivation, saying "throw away the typical image of auction."